# Current Strategies for Increasing Knock-In Efficiency in CRISPR/Cas9-Based Approaches

**DOI:** 10.3390/ijms25052456

**Published:** 2024-02-20

**Authors:** Andrés Felipe Leal, Angelica María Herreno-Pachón, Eliana Benincore-Flórez, Amali Karunathilaka, Shunji Tomatsu

**Affiliations:** 1Nemours Children’s Health, Wilmington, DE 19803, USA; lealb.af@javeriana.edu.co (A.F.L.); angelicamaria.herrenopachon1@nemours.org (A.M.H.-P.); elianapatricia.benincoreflorez@nemours.org (E.B.-F.); amali.karunathilaka@nemours.org (A.K.); 2Institute for the Study of Inborn Errors of Metabolism, Faculty of Science, Pontificia Universidad Javeriana, Bogotá 110231, Colombia; 3Faculty of Arts and Sciences, University of Delaware, Newark, DE 19716, USA; 4Department of Pediatrics, Graduate School of Medicine, Gifu University, Gifu 501-1194, Japan; 5Department of Pediatrics, Thomas Jefferson University, Philadelphia, PA 19144, USA

**Keywords:** CRISPR/Cas9, genome editing, HDR, NHEJ

## Abstract

Since its discovery in 2012, the clustered regularly interspaced short palindromic repeats (CRISPR) and CRISPR-associated protein 9 (Cas9) system has supposed a promising panorama for developing novel and highly precise genome editing-based gene therapy (GT) alternatives, leading to overcoming the challenges associated with classical GT. Classical GT aims to deliver transgenes to the cells via their random integration in the genome or episomal persistence into the nucleus through lentivirus (LV) or adeno-associated virus (AAV), respectively. Although high transgene expression efficiency is achieved by using either LV or AAV, their nature can result in severe side effects in humans. For instance, an LV (NCT03852498)- and AAV9 (NCT05514249)-based GT clinical trials for treating X-linked adrenoleukodystrophy and Duchenne Muscular Dystrophy showed the development of myelodysplastic syndrome and patient’s death, respectively. In contrast with classical GT, the CRISPR/Cas9-based genome editing requires the homologous direct repair (HDR) machinery of the cells for inserting the transgene in specific regions of the genome. This sophisticated and well-regulated process is limited in the cell cycle of mammalian cells, and in turn, the nonhomologous end-joining (NHEJ) predominates. Consequently, seeking approaches to increase HDR efficiency over NHEJ is crucial. This manuscript comprehensively reviews the current alternatives for improving the HDR for CRISPR/Cas9-based GTs.

## 1. Introduction

The clustered, regularly interspaced short palindromic repeats (CRISPR) and CRISPR-associated protein 9 (Cas9) system is a promising strategy for genome editing [1,2,3,4,5]. Since its discovery in 2012 as a genome editing tool, novel insights in several fields, such as biofuels, disease-resistant crops, novel industrial products, and targeted medicines, have arisen. Targeted medicines involve a wide range of diseases including autoimmune disorders, cancer, as well as infectious and rare diseases, among others.

The CRISPR/Cas9 system leads to precise double-strand breaks (DSB) in the DNA in an RNA-guided process [2,6,7]. Upon DSB formation, naturally, existing repair mechanisms take over, inducing indels (insertions or deletions) by activating the nonhomologous end-joining (NHEJ) pathway or the insertion of desired DNA sequences due to the homology direct repair (HDR) mechanism initiation [6,8,9]. HDR requires a DNA sequence that will be recognized by the HDR machinery and used as a template for repairing the DSB generated by Cas9 [3,9]. NHEJ is predominantly activated in mammalian cells compared to HDR [10,11]. Consequently, DBS is more susceptible to be repaired through random indels formation by the activation of the NHEJ pathway, which can be used to induce a loss of gene expression (knock-out) rather than specific DNA sequence insertions (knock-in), which is mediated by HDR. However, the CRISPR/Cas9 system is meaningfully used for correcting point mutations or inserting expression cassettes via HDR-dependent knock-in [1,12]. Knock-out has also been assessed as a therapeutical option in human diseases [13].

Even though increasing evidence supports the suitability of the CRISPR/Cas9 system for both modeling [2,14] and drug development [15,16], several challenges, such as finding alternatives for increasing the limited efficiency of the CRISPR/Cas9-mediated knock-in, are still to be overcome. For instance, genome editing in human cells mainly results in monoallelic modifications, where only a single gene copy is edited while the second copy remains unaltered or even undergoes undesired modifications due to the Cas9-mediated DSB [17]. Currently, antibiotics, or fluorescence-activated cell-sorting (FACS), are commonly used to enrich edited cells, followed by clone screening to detect biallelic modifications [17,18]. Even though these strategies work for ex vivo approaches, they are unsuitable for in vivo ones. On the other hand, it is well-known that intrinsic cell parameters (i.e., cell cycle, chromatin conformation, type of cell, etc.) can substantially decrease the genome editing efficiency; therefore, searching for modulators for those conditions could offer novel alternatives. In addition, critical design parameters regarding the CRISPR/Cas9 system (i.e., single guide RNA, donor template, delivery platform, Cas9 variant, etc.) are also pivotal for achieving higher genome editing efficiencies.

In this review, we detail the molecular mechanisms of the CRISPR/Cas9 system when using Cas9 from *Streptococcus pyogenes* (*Sp*Cas9; hereinafter referred to as Cas9) for knock-in approaches and explore the current strategies attempted to increase their efficiency in mammalian cells.

## 2. The CRISPR/Cas9 System: A Biological Perspective

Biologically, the CRISPR/Cas system is an adaptative immune response of bacteria and archaea, which protects against the invasion from viral infection and mobile elements [7,19]. After the primary invasion, a short DNA sequence is inserted within the bacterial genome at the CRISPR array. The transcription of that CRISPR array leads to the rise of a CRISPR RNA (crRNA), which interacts with a trans-encoded RNA sequence known as trans-activating CRISPR RNA (tracrRNA) [7]. The resulting interactions between crRNA and tracrRNA form a functional single guide RNA (sgRNA) that will later interact with Cas proteins. The resulting ribonucleoprotein complex (RNP) can interfere with DNA to cut it through the nuclease activity of the Cas proteins (Figure 1) [2,7].

The CRISPR/Cas systems have been classified into two major classes [20]. Class I comprises multi-Cas complexes, while class II uses a single Cas protein to mediate the DNA interference. Among them, the CRISPR/Cas9 system from *Streptococcus pyogenes* (*Sp*Cas9), belonging to class II, is the most widely studied [3,21]; nevertheless, this system also occurs naturally in several species, such as *Campylobacter jejuni*, *Francisella novicida*, *Neisseria meningitidis*, *Staphylococcus aureus*, and *Streptococcus thermophilus* [22]. The primarily described CRISPR/*Sp*Cas9 system is based on a DNA endonuclease of ~160 kDa composed of nuclease (NUC) and recognizing (REC) lobes [7,23,24]. Two catalytic domains at aspartate 10 (D10) and histidine 840 (H840) are responsible for inducing double-strand breaks (DSBs) within the DNA in a sgRNA-guided process (see Figure 1).

Currently, several modifications on Cas9 have been developed, as summarized in Table 1, and they have been modified to primarily decrease the potential off-target effect of the Cas9 proteins, although some have also been considering their fusion to key regulators of the HDR pathway to increase the knock-in efficiencies. On the other hand, an important concern in CRISPR/Cas9-based therapies is the potential Cas9-mediated unwanted DSB, which can result in genotoxic effects; thus, the use of low off-target-producing Cas9 proteins such as Cas9 nickase (nCas9), which require two RNP for inducing DSB, is often chosen over wild-type Cas9 [25]. Likewise, high fidelity Cas9 proteins, mostly containing point mutations to reduce the affinity of the Cas9 protein to the DNA, thus decreasing off-target effects, have been developed, although these genetic modifications can also decrease the on-target cutting [26]. Site-directed mutagenesis was used by Vakulskas et al. (2018) to introduce an R691A at the REC lobe. Interestingly, the authors found a lower off-target effect and retained on-target DNA cutting, as observed with wild-type Cas9 [26], providing evidence of a scarless novel Cas9 protein. Even though the selection of the particular variant will depend on the primary goal of the investigator, a variant involving high-fidelity DSB to increase HDR is highly desired. We cover high-fidelity Cas9 in Section 4.1.2 of this review paper.

## 3. The CRISPR/Cas9 System: A Genome Editing Tool

The Cas9-mediated DBS ability opened an enormous advantage for genome editing since foreign DNA can be knocked in within the genome [1]. The knock-in is currently used to recover the lack of specific gene expression that can cause human diseases, becoming a promising alternative for treating and curing thousands of disorders. In addition, the CRISPR/Cas9 system can also disrupt the expression of genes involved in several conditions via knock-out. Recently, the UK’s regulator and the Food and Drug Administration (FDA) approved Casgevy^TM^, the world’s first non-viral CRISPR/Cas9 gene editing therapy aimed to cure sickle cell disease (SCD) and transfusion-dependent β-thalassemia (TDT) [13,44]. Casgevy^TM^ targets the *BCL11A*, an erythroid-specific enhancer, in patient-derived CD34+ cells. *BCL11A* encodes for the transcription factor B-cell lymphoma/leukemia 11A, which represses the gene-encoding γ-globin. γ- and α-globin chains tetramerize to form fetal hemoglobin in erythroid cells [45]. By knocking out the *BCL11A* gene (80%), the authors found a significant increase in fetal hemoglobin synthesis [46]. Clinical trials conducted in SCD (NCT03745287) and TDT (NCT03655678) patients showed pancellularly distributed fetal hemoglobin, transfusion independence (in the TDT patient), and vaso-occlusive episode (in the SCD patient) remission.

Even though both knock-in and knock-out are feasible alternatives for treating diseases, knock-in represents a more challenging approach since inserting foreign DNA sequences requires the activation of the HDR pathway rather than the NHEJ pathway upon Cas9-mediated DSB [9,10,47]. In this regard, the cells employ different mechanisms to detect and repair DSBs. NHEJ is the predominant pathway used throughout the cell cycle except in mitosis, whereas HDR becomes the dominant pathway in the S/G2 phase instead [48]. Immediately after a DSB, several key factors in DNA damage response are activated, of which ataxia telangiectasia-mutated (ATM) protein kinase is a critical regulator activated by the MRN complex. Activated ATM phosphorylates histone variant H2AX to convert it into γH2AX. The γH2AX is recognized by MDC1, a sizeable nuclear factor that directly binds to γH2AX [49]. MDC1, in turn, recruits ATM, which leads to forming a positive feedback loop [50]. In addition, MDC1 recruits other necessary factors to the DSB sites, ultimately leading to the recruitment of BRCA1 (breast cancer type 1 susceptibility protein) and 53BP1 (p53-binding protein1). These two molecules act contrary to each other, where an accumulation of 53BP1 leads to NHEJ and BRCA1 to HDR [51]. In CRISPR/Cas9-based strategies, HDR is commonly used to insert foreign DNA sequences into the cell’s genome by delivering a donor template. This donor template can be long single-stranded oligodeoxynucleotide (lssDNA), single-stranded oligodeoxynucleotide (ssODN), double-stranded oligodeoxynucleotide (dsODN), or double-stranded DNA (dsDNA) [52], which can be either loaded into non-viral vectors or packed into viral vectors (Figure 2). In this section, we will comprehensively discuss the molecular mechanisms DSB repair mechanisms via NHEJ and HDR.

### 3.1. NHEJ Pathway

NHEJ is initiated by binding the Ku70/Ku80 (Ku) heterodimer to the DBS ends, which prevents further resection. The Ku heterodimer forms a toroid-shaped structure that can slide along DNA while scaffolding other factors [53]. After the binding of Ku, it recruits an array of factors such as DNA-dependent protein kinase catalytic subunits (DNA-PKcs), nuclease Artemis, X-ray repair cross-complementing protein 4 (XRCC4), DNA ligase IV (LigIV), XRCC4-like factor (XLF), PAXX, and Aprataxin-and-PNK like factor (APLF) [54]. The recruited DNA-PKcs bind to the Ku-DNA complex to form the DNA-PK complex. DNA-PKcs become active serine/threonine protein kinases when bound to the Ku-DNA complex. Once bound, they will translocate Ku away from the DNA ends to facilitate synapsis formation. The activated DNA-PKcs undergo autophosphorylation at several sites and phosphorylate factors such as Ku, XRCC4, LigIV, XLF, and Artemis [55].

Next, a multiprotein complex called synapsis will be formed to facilitate end-bridging. According to most studies, the key factors in synapsis formation are Ku, XRCC4, and LigIV [56,57]. To form this complex, the coiled region of XRCC interacts with the C-terminal BRCT domain of LigIV, and the BRCT of LigIV also interacts with Ku, completing the complex [58]. The XRCC4 can also act as a scaffold to recruit other NHEJ factors. The XLF and PAXX are other factors found at the synapsis, structurally similar to XRCC4. XLF can bind to XRCC4 and Ku70, further stabilizing the Ku-XRCC4-LigIV complex [59]. Furthermore, studies have found that the filament complex consists of XRCC4-XLF repeats that can wrap around the DBS ends to bridge the two ends [60]. PAXX was also found to be interacting with Ku70. Some studies suggest that XLF and PAXX may have overlapping functions [61]. Several studies found that DNA-PKcs also play a role in end bridging. However, some other studies claim that DNA-PKcs are not required for the synapsis [56].

Before ligation, the ends may need end processing depending on the type of damage caused. There could be blunt ends, overhangs, gaps, hairpins, or other structures at the damaged ends. Artemis is the major nuclease involved in end-joining, which gains 5′ and 3′ endonuclease activities after DNA-PKcs-dependent phosphorylation. Artemis usually digests the phosphodiester bond at the 5′ overhang to create a blunt end, and at the 3′ end, it leaves a short overhang of four nucleotides. In addition, there are other nucleases reported (i.e., WRN, APLF, MRN, etc.) along with enzymes such as PNKP, Apratxin, and tyrosyl-DNA phosphodiesterases 1 and 2 that can directly remove end-blocking groups to facilitate DNA end ligation [62]. The gap filling is done by the Pol X family polymerases, DNA polymerase μ (Pol μ), Pol λ, and terminal deoxynucleotidyl transferase (TdT). These enzymes can add dNTPs or rNTPs in a template-dependent or independent manner [63].

The LigIV-XRCC4-XLF complex primarily carries out the end ligation. Even though LigIV can independently perform ligation, its ligation activity is further stimulated by XRCC4, XLF, APLF, and Ku [58]. LigIV is an ATP-dependent single-turnover enzyme; therefore, it requires two copies to ligate the two strands of DNA. LigIV is the only ligase found in NHEJ and it can ligate incompatible DNA ends and ligate across gaps [64].

### 3.2. HDR Pathway

The HDR pathway repairs DSB ends by utilizing a homologous DNA template to make an accurate edit of DSB. The HDR pathway is initiated by binding the MRN (MRE11-RAD50-NBS1) complex to the DSB. This complex also acts as a scaffold to recruit other necessary proteins. Of the MRN complex, MRE11 has Mn^2+^-dependent endonuclease activity, the RAD50 homodimer forms the core of the complex, which also holds DSB ends together, and NBSI further recruits ATM [11].

Next, the C-terminal binding protein interacting protein (CtIP) nuclease is recruited; together, these factors promote short-range end resection where a short 3′ ssDNA overhang (15–20 bp) is made. However, for resection to occur, CtIP first needs to be phosphorylated by CDK. This process is also promoted by additional factors such as BRCA1-BARD1. The short overhangs allow the binding of exonuclease 1 (EXO1) and/or the DNA replication ATP-dependent helicase/nuclease DNA2 (Dna2)/bloom syndrome protein (BLM) complex that further extends the 3′ overhangs leading to long-range resection (2–4 kb) [65,66]. EXO1 has 5′ to 3′ exonuclease activity. Of the BLM/DNA2 complex, BLM facilitates the separation of DNA strands, which allows DNA2, a 5′ and 3′ endonuclease, to create long ssDNA [66]. The exposed long 3′ overhang is immediately stabilized by the rapid binding of replication protein A (RPA) [67].

The RPA is then replaced by RAD51, which forms a nucleoprotein presynaptic filament; this process is stimulated by breast cancer 1 (BRCA1), BRCA2, and the partner and localizer of BRCA2 (PALB2) along with some other factors [68,69]. RAD51 is a DNA strand-exchange protein that promotes the 3′ nucleofilament-mediated homology searches and strand invasion. In this process, one end of the 3′-overhang is aligned with a homologous DNA strand to form a displacement loop (D-loop) [70]. The 3′-overhang extends on the homologous DNA strand with the aid of proliferating cell nuclear antigen (PCNA), DNA polymerase δ (Polδ), and the clamp loader complex RFC1–5 to synthesize the complementary strand [71,72]. Finally, the resolution of the exchanged DNA strands can occur through at least three different pathways: the double Holliday junction (dHJ) crossover and non-crossover pathway, non-crossover synthesis-dependent DNA strand annealing (SDSA), and break-induced replication (BIR) [71].

## 4. Strategies for Increasing CRISPR/Cas9-Mediated HDR Efficiency

As we have detailed, NHEJ is a more predominant DSB repair mechanism than HDR in eukaryotic cells. Consequently, increasing the HDR pathway by either blocking NHEJ or enhancing HDR itself is critical for successful CRISPR/Cas9-mediated knock-in. In this regard, factors such as the design of the CRISPR/Cas9 system, the type of cell to edit, and the molecular DSB repair pathways’ modulation have been recognized as key factors for enhancing CRISPR/Cas9-mediated DNA insertion efficiency (Figure 3). These critical considerations will be covered in this section.

### 4.1. CRISPR/Cas9 Design Considerations

#### 4.1.1. Cas9 Delivery Platforms

The CRISPR/Cas9 comprises a sgRNA in combination with the Cas9 endonuclease, which interacts with the targeted DNA to mediate DSB [7]. Currently, these components can be delivered to the cell as DNA, mRNA, or as an in vitro preformed RNP complex [2,73], which can affect the on-target DNA cutting, thus exerting different knock-in efficiencies. In this regard, Kouranova et al. (2016) evaluated the efficacy of DNA, mRNA, and RNP for knocking out ApoE, Ptsg1, and Rosa26 locus in mammalian cells. Interestingly, the delivery of Cas9 mRNA or Cas9 plasmid and sgRNA showed low or non-on-target cleavage. In contrast, the RNP complex exhibited higher efficacy in all loci assessed, as demonstrated by the in vitro cleavage assay, although the percentage of indels formation was not calculated [74]. It is important to note that, even though RNP are traditionally preferred over plasmids (DNA) or mRNA, especially when working with primary cells, purified Cas9 proteins are costly and are an important limitation for many laboratories. Therefore, plasmids carrying both Cas9 sequences and mRNA as an all-in-one plasmid are also feasible. All-in-one plasmids have shown high DSB at the AAVS1 [75] and Rosa26 [76] locus in mammalian cells, supporting the idea that low-cost alternatives can be successfully implemented.

#### 4.1.2. Cas9 Fusion Variants

The fusion of Cas9 proteins to motifs able to modulate the HDR pathway is a well-known strategy for increasing HDR events upon Cas9 DBS [11,27]. Recently, Ma et al. (2020) developed a Cas9 variant termed miCas9, which fuses Brex27 to Cas9 [27]. Brex27 is a 36 amino acid motif interacting with RAD51 to stabilize RAD51/ssDNA nucleoprotein filaments. miCas9 showed higher efficiency rates (2–3-fold increase) for large-size gene knock-in compared to wtCas9, while significantly decreasing off-target events [77]. Even though other HDR-fusion motifs, such as CtIP [30], DN1S [31], Geminin [78], mSA [79], and RAD52 [80], have also been evaluated, these motifs are larger than Brex27 (~306% and 1308%), resulting in potential challenges when viral vectors are used as carriers for the CRISPR/Cas9 system due to their limited packing capacity [81,82]. Even though some efforts have been made to increase the Cas9 specificity without size changes resulting in several Cas9 variants (see Table 1), they have a marginal effect on the knock-in efficiency increase [26,27,33].

On the other hand, it has also been reported that cell cycle synchronization and the timing of Cas9 delivery can enhance the HDR pathway [83], supporting the idea that the synchronization of Cas9 expression to the cell cycle can lead to higher HDR events. In this regard, Gutschner et al. (2016) showed that the fusion of Cas9 to the N-terminal segment of human Geminin not only confers nuclear Cas9 localization, but also increases the HDR efficiency by 87% when compared to wtCas9 [29]. Moreover, Nocodazole-mediated chemical M phase cell cycle inhibition has shown to be a potent enhancer of HDR in HEK-293T cells [83], supporting that both alternatives, Cas9 fusion variants, and chemical cell cycle synchronization can synergically increase HDR events.

Likewise, the recent discovery of anti-CRISPR (Acr) proteins, inhibitors of the CRISPR-Cas9 system, offers the potential to control Cas9 activity based on the cell cycle [84]. By fusing AcrIIA4 with the N-terminal region of the human chromatin licensing and DNA replication factor 1 (hCdt1), Matsumoto et al. (2020) demonstrated that Cas9 activity can be activated during the S/G2 phases and inactivated during the G1 phase, thus offering a cell cycle-dependent Cas9 activity system [85]. This innovative approach effectively reduced NHEJ-induced mutagenesis and off-target effects while boosting the efficiency of HDR. It is particularly advantageous when using single-stranded oligodeoxynucleotides (ssODN) as templates for HDR, enhancing the HDR/NHEJ ratio compared to wtCas9. This innovative system is expected to have broad applications across various CRISPR-Cas systems.

CtlP, a crucial protein in the early stage of HDR, has been harnessed to stimulate HDR efficiency. The fusion of Cas9 with both full-length CtIP and a minimal N-terminal fragment of CtlP facilitated its entry to the DNA cleavage site, resulting in a 2-fold or greater increase in HDR efficiency. This approach was demonstrated in various cell types, including human fibroblasts, iPS cells, and rat zygotes. However, it is essential to note that the patterns of insertions and deletion induced by the modified Cas9 differed from those generated by wild-type Cas9, suggesting a different balance of repair pathways [30]. It is notable that a novel fusion Cas9 protein comprised of eRad18 and CtlP, termed Cas9-RC, was recently tested for increasing HDR through its intra-uterus administration in embryonic mouse brains [86] for inserting an expression cassette carrying mCherry at the Actβ gene. Cas9-RC led to a significant increase of an up to 3.7-fold knock-in increase compared to the non-fused Cas9 protein. Despite these promising findings, Cas9-RC did not show enhanced knock-in efficiency as compared to Cas9 at the Negr1 locus [86], suggesting that either internal chromatin aspects on the targeted locus and large size Cas9 variants can limit the efficiency of the CRISPR/Cas9 system.

Subsequently, Cas9 was fused with other HDR-related proteins like Mre11 and Rad52, showing similar improvements in human HEK-293T cells [87] and yeast cells [88]. Similarly, the yeast RAD52 (yRAD52) and its fusion with Cas9 have been proposed as a promising alternative for enhancing HDR efficiency. In contrast with human RAD52 (hRAD52), yRAD52 showed a backup hRAD52 function for the BRCA1-BRCA2 pathway of HDR [89]. Based on that knowledge, Shao et al. (2017) evaluated the yRAD52 and Cas9 co-expression and the yRAD52-Cas9 fusion approach, which yielded an up to 3.8-fold increase in HDR during various reporter assays and genome editing experiments in both HEK-293T and porcine PK15 cells. Furthermore, yRAD52, combined with the NHEJ inhibitor SCR7, led to a remarkable increase in HDR efficiency. This combined strategy proved effective in modifying the *IGF2* gene in porcine cells, demonstrating a 2.2-fold rise in the HDR frequency [90].

#### 4.1.3. sgRNA

Designing sgRNA is a critical step for successfully Cas9-mediated DNA cutting. Thus, choosing good sgRNA candidates is critical to favor the interactions between Cas9 and the targeted DNA through base complementary, achieving high on-target cutting [7,30], which is critical for activating repair mechanisms to induce HDR. Several predictors are available for the in-silico evaluation of potential candidates in several genome species and different Cas proteins [91,92]. Online predictors score sgRNA according to their on-target activity based on sophisticated studies published by Doench et al. (2016) [93]. Parameters scoring sgRNA include the position and frequency of single and dinucleotides, the content of GCs across the sgRNA, and melting temperatures. The Doench’s rules also determine the cutting frequency of a sRNA to evaluate the off-target sites. For pairing DNA nicking when using nCas9, the predictors also provided information regarding PAM configuration (out and in) and sgRNA comprising several off-sets. PAM-out configuration, together with off-sets between 40–70 bp and 50–70 bp when nCas9 (D10A) or nCas9 (H840A) are used, respectively, have demonstrated higher on-target cleavage efficiency [39], while decreasing between 50–1500-fold the Cas9 off-target events in comparison to wtCas9 [39,94].

#### 4.1.4. Donor Template

Both dsDNA and ssDNA can be donor templates for CRISPR/Cas9-mediated insertions [95]. Donor templates comprise two homologous recombination arms (HRA) flanking a sequence that will be inserted into the genome, ranging from a single nucleotide (nt) to thousands of them. Conventionally, dsDNA donor templates are preferred to insert larger than 100 nt sequences, while ssDNA is commonly used when less than 100 nt needs to be inserted [95]. Although symmetric HRAs in the donor templates have been successfully assessed in several genomic regions for inserting foreign DNA sequences [75,96], asymmetric HRAs are also plausible. They can even improve knock-in efficiencies when longer 5′ arms are used [97,98]. A comprehensive review in this regard was recently published by Shakirova et al. (2023) [52].

### 4.2. Blocking the NHEJ Pathway

#### 4.2.1. NHEJ Starting Pathway

In the G1 phase of the cell cycle, the 53BP1 suppressed the DNA end resection by binding the DSB and blocking the initiation of the HDR [71,99,100,101]. Consequently, blocking 53BP1 would promote the HDR pathway. Under this presumption, Canny et al. (2018) tested the effect of the i53 (an inhibitor of 53BP1) on the HDR rate in HEK-293T and K562 cells [102]. As expected, an HDR improvement of 1.3-fold and 1.8-fold was noticed for 293T and K562 cells, respectively, supporting the idea that 53BP1 blocking can be an exciting alternative for increasing CRISPR/Cas9-based genome editing events. Similarly, the effect of i53 was tested in CD34+ cells, showing a significant HDR increase of 2.3-fold compared to the absence of i53 [103].

#### 4.2.2. Initial DSB Recognition by DNA-PKcs

Given the critical involvement of the DNA-PKcs in the phosphorylation of multiple factors during the NHEJ pathway, DNA-PKcs inhibition has also been widely explored via small molecules such as NU7026, Ku-0060648, M3814, and NU7441 in several cell models including HEK-293T, CD34+, and iPSCs, among others [99,100,101]. Even though these molecules can enhance the HDR between 1.7- to 10-fold [99,100,101,104], their effect seems cell-dependent. For instance, u7026 and NU7441 did not show the same impact in the HDR increase in mouse embryonic stem cells [100,105,106]. Likewise, the use of M3814 in iPSCs and T cells led to a 2.8-fold HDR increase [107,108,109], while small molecules such as SCR7, well-known for enhancing CRISPR/Cas9-related HDR in tumoral cells [110], did not improve the HDR efficiency in iPSCs and T cells [107]. Most recently, Selvaraj et al. (2023) screened out six small molecules that inhibit DNA-PKcs in primary cell lines [111]. In this study, AZD7648 was identified as a booster of HDR, achieving an up to 50-fold CRISPR/Cas9-mediated knock-in events in primary cells. These interesting findings strongly support evaluating the potential HDR enhancers in several cell lines.

#### 4.2.3. DNA Ligase IV-Dependent Ligation

Scr7, an inhibitor of ligase IV, has been tested in several human cells (HEK-293T cells and iPSCs) and murine models (mouse embryonic stem cells and Murine zygotes), among other species, showing a CRISPR/Cas9-mediated HDR efficiency increase ranging between 1- to 19-fold [99,100,101,112]. Scr7 has also improved the efficiency of CRISPR/Cas9 in targeting multiple loci in mammalian cells and mouse zygotes [113]. Recently, Tanihara et al. (2023) tested the suitability of Scr7 for introducing a point mutation into the porcine insulin (*INS*) gene on porcine zygotes. Interestingly, the authors found no evident improvement when several Scr7 concentrations (0.5 to 4 µM) were tested [114], suggesting a limited effect of Scr7 on the HDR efficiency improvement. On the other hand, Chu et al. (2015) targeted DNA ligase IV for increasing CRISPR/Cas9-mediated HDR [115]. In this approach, authors co-expressed the E1B55K and E4orf6, two viral proteins from adenovirus able to activate the ubiquitin-proteosome pathway [116], together with the Cas9 protein achieving an up to 8-fold HDR increase in human and mouse cell lines.

#### 4.2.4. Small Interference (si) and Short Harping (sh) RNA NHEJ Protein Downregulation

Some studies have demonstrated the feasibility of downregulating the NHEJ pathway’s critical components, leading to increased HDR [11]. In this scenario, the siRNA-mediated downregulation of the Ku70 and Ku80 complex showed an improvement of up to 3-fold in HDR efficiency in pig embryonic fibroblasts [117,118]. Similarly, shRNAs against Ku70 and Ku80 have shown increased HDR events up to 14% in mammalian cells [115]. Even though these alternatives show novel options that can be attempted to improve HDR, several challenges, including the persistence of NHEJ pathway inactivation, are still a concern [11,117]. In addition, a novel strategy by co-expressing Cas9 and the micro-RNA 21 (miRNA-21) led to a significant increase in the knock-in of SOX2 in iPSCs, reaching up to 3-fold as compared with the knock-in efficiency in the absence of miRNA-21 [8]. miRNA is a well-known modulator of several proapoptotic genes, such as caspase 3 [119,120].

### 4.3. Stimulating the HDR Pathway

As mentioned before, the frequency of HDR repair is lower than NHEJ in mammalian cells; thus, blocking the NHEJ pathway has been the primary goal for increasing CRISPR/Cas9-mediated HDR events. Nevertheless, the direct modulation of HDR has also been addressed by modifying Cas9 proteins (see Section 4.1.2). This last section covers strategies based on non-Cas9 direct modifications nor blocking the NHEJ pathway.

#### 4.3.1. Small Molecules

The small molecule RAD51-stimulatory compound 1 (RS-1), which stabilizes RAD51 [100], was assessed as an HDR enhancer [121,122]. A study in rabbit embryos treated with RS-1 showed a significant CRISPR/Cas9-mediated increase in the HDR events up to 2- to 6-fold at different loci [122]. Similarly, RS-1 increased HDR rates from 1.5- to 6-fold in mammalian cells, including K562, HEK-293T, iPSCs, bovine zygotes, and zebrafish embryos [101], supporting its use across several primary cells and cell lines.

#### 4.3.2. Modulating Chromatin Remodeling Factors

The arresting of the cell cycle and DNA accessibility by regulating chromatin remodeling factors like histone acetylation has been studied as an alternative approach [99]. In the acetylation of the histone proteins, chromatin is remodeled, exposing the DNA strand to active regulation, a state known as euchromatin. In contrast, when the histone tails are deacetylases, the chromatin is condensed, and the DNA strand is negatively regulated, a state known as heterochromatin [123]. Consequently, the inhibition of histone deacetylases (HDACs), enzymes that catalyze the removal of acetyl groups from histone proteins [124,125,126], should increase the HDR as long as the chromatin becomes euchromatin. This premise has been tested for several HDAC inhibitors (HDACis), such as trichostatin A (TSA), PCI-2478, and vorinostat, among others [126,127]. For instance, in iPSC cells, TSA can increase CRISPR/Cas9-mediated HDR events by up to 4-fold compared to the efficiency achieved in the absence of this HDACi [127] by arresting iPSC cells in the G2/M phase cell cycle [124]. These findings strongly suggest that chromatin modulation could also be a promising strategy for increasing CRISPR/Cas9-mediated HDR events.

## 5. Future Perspectives

There are no doubts about the promise of CRISPR/Cas9-based therapies. The recent approval in Europe and the USA of Casgevy^TM^, the first CRISPR/Cas9-based FDA-approved gene therapy, is hopefully the first of hundreds of innovative strategies using the CRISPR/Cas9 system. Nevertheless, increasing knock-in efficiency is still a high priority, given the low frequency of the HDR in mammalian cells. So far, several alternatives, including primarily the inhibition of the NHEJ pathway, have shown enhanced knock-in efficiencies (up to 50-fold) compared to classical genome editing in which no NHEJ inhibitors are included. For instance, the recent identification of the AZD7648, a well-known DNA-PK inhibitor, as an HDR booster in primary cells could offer a promising alternative for ex vivo CRISPR/Cas9-based gene therapies. The modulation of the HDR pathway has been less explored through small molecules and chromatin remodeling factors, which could also offer novel alternatives to increase HDR-mediated gene insertion. In Table 2, we summarized the outcomes of the strategies attempted to boost HDR.

As shown in Table 2, several studies point out cell dependence, suggesting that cell physiology influences the successful outcome of the CRISPR/Cas9 system in combination with direct or indirect HDR stimulators. Consequently, we highlight the need to explore in more detail the current alternatives for achieving high knock-in efficiencies by using either NHEJ inhibitors or HDR stimulation in cell-specific contexts, together with deep cell analysis to uncover not only the mechanism behind improved genome editing, but also the short- and long-term cellular consequences of the modulation of such vital cell pathways. For instance, NHEJ is a well-conserved pathway responsible for generating immunoglobin class switch recombination where key components such as Ku70/Ku80, DNA ligase IV, XRCC4, γH2AX, 53BP1, MDC1, and Mre11–Rad50–NBS1 are involved [128,129]; therefore, assessing the impact on the downregulation of these NHEJ components by sh- or si-RNA relies on fundamental concerns that should be address. Likewise, the loss of some NHEJ has been associated with genome instability [130]. Therefore, the potential genotoxic effect of NHEJ protein modulation remains to be uncovered.

## Figures and Tables

**Figure 1 ijms-25-02456-f001:**
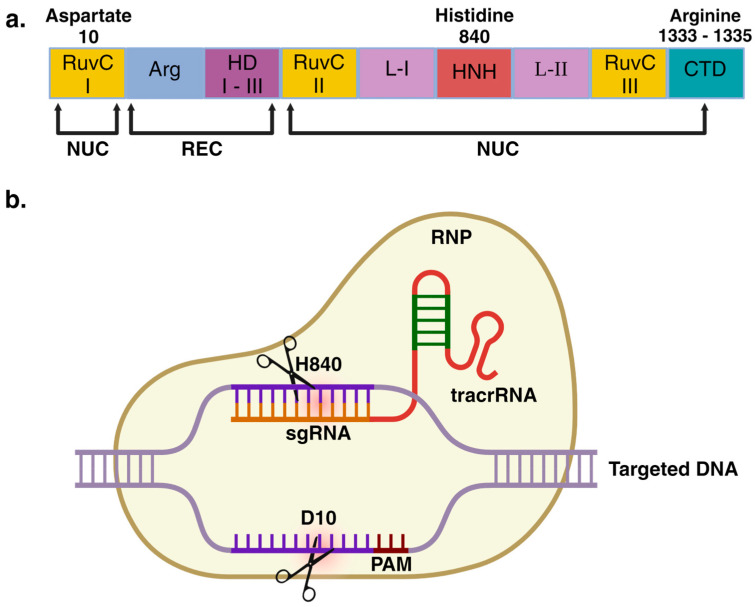
Cas9 protein structure and Cas9-mediated DNA double-strand breaks. In (**a**), the full Cas9 protein structure isolated from *Streptococcus pyogenes* is shown. Notice the recognizing (REC) and nuclease (NUC) lobes, which mediate the targeted DNA recognition and double-strand break, respectively. The arginine (Arg) 1333 and 1335 residues responsible for PAM sequence recognition are also displayed. In (**b**), a classical overview of the ribonucleoprotein (RNP) complex (Cas9 together with sgRNA) interacting with the targeted DNA is shown. The histidine 840 (H840) catalyzes the break of the sgRNA interacting DNA strand, while aspartate 10 (D10) mediates breaking on the opposite strand. Note that the RuvC endonuclease domain comprises three segments: RuvC I, II, and III, while the HNH catalytic domain comprises only one segment. Likewise, three alpha-helical domains (HD) at the REC lobe are placed and primarily responsible for nucleic acid binding. Finally, L-I and L-II are key linkers that aid the connection between RuvC and HNH. CTD: Carboxy-terminal domain. This figure was created with BioRender.com.

**Figure 2 ijms-25-02456-f002:**
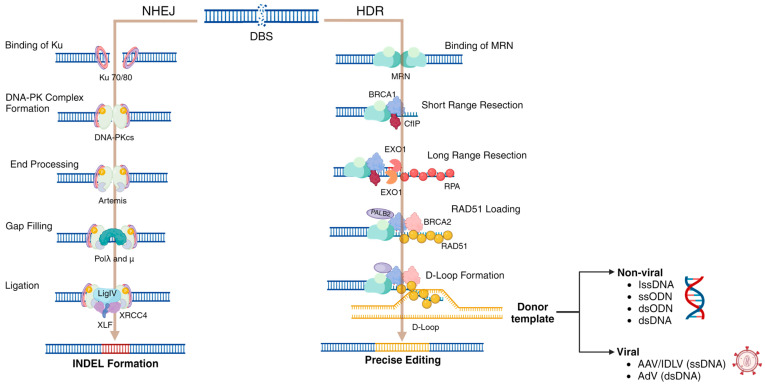
Major steps in non-homologous end joining (NHEJ) and Homology-directed repair (HDR). NHEJ is initiated by binding of Ku to the DSB ends. This is followed by recruiting DNA-PKcs and other necessary scaffolding factors that bring DSB ends together. Nucleases process the incompatible ends, polymerases fill gaps, and finally, the ends are ligated by DNA ligases. This process ultimately leads to the formation of INDELs. HDR is initiated by the binding of MRN that stabilizes DSB ends. Then, 5′ exonucleases produce short-range resections followed by long-range resections. The long 3′ overhangs are directed for homology search and strand invasion by RAD51, leading to D-loops’ formation. Finally, the exchanged DNA strands are resolved, leading to precise editing. Several platforms can be used as donor templates, including long single-stranded oligodeoxynucleotide (lssDNA), single-stranded oligodeoxynucleotide (ssODN), double-stranded oligodeoxynucleotide (dsODN), and double-stranded DNA (dsDNA) which can be carried through non-viral vectors, as well as DNA templates loaded into viral vectors. For more details related to donor templates and their impact on HDR, we strongly encourage readers to review the paper by Shakirova et al., 2023 [52]. Although differences in the recognition of ssDNA (i.e., BRCA1) and dsDNA (i.e., BRCA2) have been well-documented, in both scenarios, the homologous recombination takes place, leading to precise genome editing in the context of the CRISPR/Cas9 system. This figure was created with BioRender.com.

**Figure 3 ijms-25-02456-f003:**
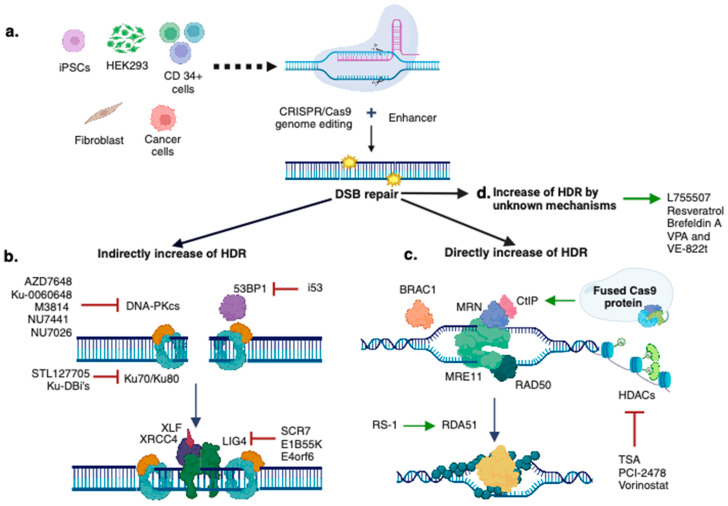
Strategies for increasing the knock-in efficiency in CRISPR/Cas9-based genome editing. Increasing CRISPR/Cas9-mediated genome editing events involves factors such as the type of cell (**a**), as well as indirect (**b**) or direct (**c**) HDR modulation by using several strategies. Several small molecules can also achieve HDR increase, although their full mechanism of action is still to be uncovered (**d**). This figure was created with BioRender.com.

**Table 1 ijms-25-02456-t001:** Cas9 variants and their major characteristics.

Variant	Type of DNA Cutting	Characteristics	Advantages	Limitations	Ref.
wtCas9	DSB	Induces DSB	High DBS efficiency Simplest sgRNA design	High off-target	[7]
HF-Cas9	DBS	High-fidelity	Low off-target	* Reduced on-target cut	[26,27,28]
HDR-Cas9	DBS	Fusion of several motifs to interact with HDR protein	Increases HDR	** Large in-size proteins upon fusion	[27,29,30,31]
eCas9	DSB	Presence of 4X NLS	High-fidelity High nuclear localization	Reduced on-target cut	[32,33]
xCas9	DSB	Recognizes several PAMs, NGG, NG, NNG, GAT, and CAA	Broader PAM recognition Low off-target	Altered interactions between Cas9 and sgRNA/DNA duplex	[34,35,36]
HA-Cas9	DSB	High fidelity	Higher DSB cutting as compared to wtCas9	Moderate off-target	[37,38]
nCas9	SSB	HNH or RuvC mutated Requires paired sgRNA for DSB cutting	High DSB efficiency	Lowest off-target Require accurate sgRNA design	[39]
dCas9	NA	HNH and RuvC mutated No DNA cutting	Allows transcriptional studies without permanent modification in the DNA	Large-in-size proteins Moderate off-target	[40,41]
CD-Cas9	NA	Conditional Cas9 expression in the presence of FKBP12	Temporal control of Cas9 Marginal off-target effect	Large in-size protein Background mRNA expression	[42,43]

* The on-target is unaffected when modification refers to R691A [26]. ** A novel Cas9 protein called miCas9 which carries the Brex27 motif results in the smallest HDR-Cas9 fusion protein described so far [27]. wtCas9: Wild-type Cas9; HF-Cas9: High-fidelity Cas9; miCas9: Cas9 fused to Brex27; NLS: Nuclear Location signals; xCas9: Cas9 with expanded PAM recognition; HA-Cas9: Hyper-accurate Cas9; nCas9: nickase Cas9; dCas9: Dead Cas9; CD-Cas9: Conditional deactivated Cas9; DSB: Double-strand break; SSB: Single-strand break; NA: Not applicable.

**Table 2 ijms-25-02456-t002:** Outcome of NHEJ and HDR modulators for boosting HDR in mammalian cells.

Strategy	Locus	* Cas9 Variant	Cells	** % HDR Efficiency (Fold-Change)	Ref.
Cas9-Brex27 (miCas9)	AAVS1	wtCas9	Fibroblast	2.9% (2.5-fold)	[27]
	AAVS1	wtCas9	AECs	1.5% (2.1-fold)	
	AAVS1	wtCas9	iPSCs	1.2% (2.1-fold)	
CtIP fusion to Cas9	AAVS1	wtCas9	Fibroblast	2.3-fold	[30]
	AAVS1	wtCas9	HEK293	2.4-fold	
	ATF4	wtCas9	HEK293	2.5-fold	
	GABP	wtCas9	HEK293	1-fold	
	TGIF2	wtCas9	HEK293	2.1-fold	
	RAD21	wtCas9	HEK293	2.5-fold	
	CREB	wtCas9	HEK293	1.4-fold	
	AAVS1	wtCas9	iPSCs	1.5-fold	
Cas9-RC	Actβ	wtCas9	Neuron P7	3.7-fold	[86]
DN1S fusion to Cas9	AAVS1	wtCas9	HEK293	33.3% (1.6-fold)	[31]
	LMO2	wtCas9	HEK293	54.6%(2-fold)	
	CD45	wtCas9	K562	17% (1.3-fold)	
	CCR5	wtCas9	Jurkat	26% (1.1-fold)	
Geminin fusion to Cas9	MALAT1	wtCas9	HEK293	14% (1.4-fold)	[29]
AcrIIA4-hCdt1 co-expressed with Cas9	AAVS1	wtCas9	HEK293	1.7-fold	[85]
	EMX1	wtCas9	HEK293	4-fold	
	VEGFA	wtCas9	HEK293	4.5-fold	
yRAD52-Cas9 fusion protein	VEGF	wtCas9	HEK293	3.3-fold	[90]
	CCR5	wtCas9	HEK293	3.8-fold	
	IGF2	wtCas9	PK15	2.2-fold	
i53	LMNA	wtCas9	U2OS	8.6% (1.8-fold)	[102]
	HIST1H2BK	wtCas9	HEK293	2.1% (1.3-fold)	
	HIST1H2BK	wtCas9	K562	12% (1.8-fold)	
	CYBB	wtCas9	CD34+	2.3-fold	[103]
NU7026	CALD1	wtCas9	iPSC	25.5% (1.5-fold)	[104]
	CALD1	nCas9	iPSC	11.7% (1.5-fold)	
	KATNA1	wtCas9	iPSC	5.2% (1.6-fold)	
	KATNA1	nCas9	iPSC	18.7% (2.6-fold)	
	SLITRK1	wtCas9	iPSC	6.2% (1.2-fold)	
	SLITRK1	nCas9	iPSC	11.9% (2.5-fold)	
Trichostatin A	EEF1A1	wtCas9	iPSCs	30% (2.2-fold)	[107]
	EEF2	wtCas9	T cells	50% (1.3-fold)	
Trichostatin A	CALD1	wtCas9	iPSCs	12.3% (0.8-fold)	[104]
	CALD1	nCas9	iPSCs	10.1% (1.5-fold)	
	KATNA1	wtCas9	iPSCs	3.4% (1-fold)	
	KATNA1	nCas9	iPSCs	17.6% (2.2-fold)	
	SLITRK1	wtCas9	iPSCs	5% (1-fold	
	SLITRK1	nCas9	iPSCs	8.7% (1.8-fold)	
RS-1	CALD1	wtCas9	iPSCs	22.2% (1-fold)	[104]
	CALD1	nCas9	iPSCs	8.5% (1.1-fold)	
	KATNA1	wtCas9	iPSCs	2.5% (0.8-fold)	
	KATNA1	nCas9	iPSCs	8% (1.2-fold)	
	SLITRK1	wtCas9	iPSCs	4.4% (0.9-fold)	
	SLITRK1	nCas9	iPSCs	5.3% (1-fold)	
Resveratrol	CALD1	wtCas9	iPSCs	13.3% (1.1-fold)	[104]
	CALD1	nCas9	iPSCs	7.1% (1-fold)	
	KATNA1	wtCas9	iPSCs	2.7% (0.8-fold)	
	KATNA1	nCas9	iPSCs	8% (1.1-fold)	
	SLITRK1	wtCas9	iPSCs	4.3% (0.9-fold)	
	SLITRK1	nCas9	iPSCs	6.2% (0.9-fold)	
SCR7	TSG101	wtCas9	MelJuSo	19-fold	[113]
	Tap1	wtCas9	DC2.4	58.3% (13-fold)	[113]
	AAVS1	wtCas9	MCF-7	4% (3-fold)	[110]
	AAVS1	wtCas9	HCT-116	3.1% (2.4-fold)	
	CALD1	wtCas9	iPSCs	12.5% (0.9-fold)	[104]
	CALD1	nCas9	iPSCs	7.1% (1.1-fold)	
	KATNA1	wtCas9	iPSCs	2.8% (0.8-fold)	
	KATNA1	nCas9	iPSCs	6.3% (1-fold)	
	SLITRK1	wtCas9	iPSCs	4.7% (0.9-fold)	
	SLITRK1	nCas9	iPSCs	5.9% (1.1-fold)	
L755507	CALD1	wtCas9	iPSCs	11.6% (0.9-fold)	[104]
	CALD1	nCas9	iPSCs	5.5% (1-fold)	
	KATNA1	wtCas9	iPSCs	3.4% (0.9-fold)	
	KATNA1	nCas9	iPSCs	5.6% (1-fold)	
	SLITRK1	wtCas9	iPSCs	3.9% (0.7-fold)	
	SLITRK1	nCas9	iPSCs	4.6% (0.9-fold)	
STL127685	CALD1	wtCas9	iPSCs	12.8% (1-fold)	[104]
	CALD1	nCas9	iPSCs	7.4% (0.9-fold)	
	KATNA1	wtCas9	iPSCs	2.7% (0.8-fold)	
	KATNA1	nCas9	iPSCs	6.4% (0.9-fold)	
	SLITRK1	wtCas9	iPSCs	4.9% (1-fold)	
	SLITRK1	nCas9	iPSCs	4.9% (0.9-fold)	
M3814	EEF1A1	wtCas9	iPSCs	40% (2.9-fold)	[107]
	EEF2	wtCas9	T cells	70% (2-fold)	
	LAG3	wtCas9	T cells	2.8-fold	[109]
AZD7648	CCR5	wtCas9	CD34+	3-fold	[111]
	CCR5	wtCas9	iPSCs	8.5-fold	
	CCR5	wtCas9	T cells	7-fold	
	HBB	wtCas9	CD34+	2.6-fold	
	STING1	wtCas9	CD34+	2.3-fold	
	STING1	wtCas9	iPSCs	50-fold	
	STING1	wtCas9	CD34+	6-fod	
E1B55K and E4orf6	Rosa26	wtCas9	NIH3T3	8-fold	[115]
siRNA targeting Ku70/Ku80	B-actin	wtCas9	PFF	5.6-fold	[117]
	H11	wtCas9	PFF	1.9-fold	
shRNA targeting Ku70	Rosa26	wtCas9	NIH3T3	2.1-fold	[115]
shRNA targeting LIG4	Rosa26	wtCas9	NIH3T3	2.8-fold	
shRNA targeting Ku70/Ku80	Rosa26	wtCas9	NIH3T3	3.0-fold	
shRNA targeting Ku80/LIG4	Rosa26	wtCas9	NIH3T3	3.8-fold	
shRNA targeting Ku70/LIG4	Rosa26	wtCas9	NIH3T3	5-fold	
miRNA-21	SOX2	wtCas9	iPSCs	3-fold	[8]
RS-1	RLL	wtCas9	R-Em	26.1% (6-fold)	[122]
	CFTR	wtCas9	R-Em	30% (2.4-fold)	
Trichostatin A	HIST1H2BJ	wtCas9	iPSCs	4-fold	[127]

* Cas9 variants are referred as wild-type (wtCas9; RuvC and HNH active domains) or Cas9 nickase (nCas9; missing one catalytic domain). ** In some cases, fold change was determined by calculating the ratio of the positive cells carrying the desired knock-in (i.e., expressing a fluorescent marker, a restriction enzyme-based evaluation, etc.) when edited in the presence or absence of the HDR enhancer. The table presents the highest value reported by authors under the above-described conditions. AECs: Airway epithelial cells; iPSCs: Induced pluripotent stem cells; HEK293: Human embryonic kidney cells; K562: Human myelogenous leukemia cells; PK15: Pig kidney cells; U2OS: Human osteosarcoma; CD34+: Hematopoietic stem cells; DC2.4: Murine dendritic cells; MCF-7: Human breast cancer cells; HCT-116: Human colorectal carcinoma cells; NIH3T3: Mouse embryonic fibroblasts; PFF: Pig fetal fibroblasts; R-Em: Rabbit embryos.

## Data Availability

Not applicable.

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
