# Peer review of "Current Strategies for Increasing Knock-In Efficiency in CRISPR/Cas9-Based Approaches"

_ijms, 2024, doi:10.3390/ijms25052456_

Round 1
Reviewer 1 Report
Comments and Suggestions for Authors
The manuscript submitted by Andrés Felipe Leal et al. reviews the current strategies to increase HDR in different cell types and promote knock-in efficiency in CRISPR/Cas9-based approaches. The goal is to open new avenues in gene therapy. The main data are described in this review. However, some sections lack a sufficient level of detail (such as cell type effect, donor effect, delivery, Cas9…) while others are perhaps too detailed in comparison (e.g., description of the NHEJ pathway). The authors need to complete some points in this review.
Page 62: Intrinsic cell parameters should be further discussed in this review. The efficiency of KI will depend on the cell type, delivery platform, donor template design, etc. Could the authors provide more details on these different points in the text?
Figure 2: The figure needs to be more described in the legend. What do HD, HNH, etc., stand for? It is unclear.
Table 1: The use of Cas9 is crucial in KI approaches. The table is incomplete and needs to be further described in the text. What are the advantages for each Cas9? Which Cas9 is the best for KI? miCas9 is not included in the table.
NHEJ paragraph: There are too many details that are difficult to follow. Is it necessary to understand the current strategies used to improve KI?
4.1.2: What is the goal of this paragraph? Can the design of RNA guides improve the KI? It is not clear.
4.1.3. The design of the donor is important to improve KI (Size of arms, size of the donor…). It lacks some details in this review.
Cell type effect: It would be appreciated to have a table of the best strategies to improve HDR efficiency for each cell type.
Conclusion: It would be useful to comment on the disadvantages of some approaches to improve HDR. What are the other consequences on the genome when an NHEJ inhibitor is used?
Author Response
Reviewer 1. The manuscript submitted by Andrés Felipe Leal et al. reviews the current strategies to increase HDR in different cell types and promote knock-in efficiency in CRISPR/Cas9-based approaches. The goal is to open new avenues in gene therapy. The primary data are described in this review. However, some sections lack a sufficient level of detail (such as cell type effect, donor effect, delivery, Cas9…) while others are perhaps too detailed in comparison (e.g., description of the NHEJ pathway). The authors need to complete some points in this review.
- Page 62: Intrinsic cell parameters should be further discussed in this review. The efficiency of KI will depend on the cell type, delivery platform, donor template design, etc. Could the authors provide more details on these different points in the text?
Answer. We appreciate this comment, which refers to the introduction section of our review paper. We have extended this discussion in numeral 4.1.1, including a new section entitled Cas9 delivery platforms as follows:
4.1.1 Cas9 delivery platforms. The CRISPR/Cas9 comprises a sgRNA in combination with the Cas9 endonuclease, which interacts with the targeted DNA to mediate DSB [7]. Currently, these components can be delivered to the cell as DNA, mRNA, or as an in vitro preformed RNP complex [2, 73], which can affect the on-target DNA cutting, thus exerting different knock-in efficiencies. In this regard, Kouranova et al. (2016) evaluated the efficacy of DNA, mRNA, and RNP for knocking out ApoE, Ptsg1, and Rosa26 locus in mammalian cells. Interestingly, delivery of Cas9 mRNA or Cas9 plasmid and sgRNA showed low or non-on-target cleavage. In contrast, the RNP complex exhibited higher efficacy in all lo-ci assessed, as demonstrated by in vitro cleavage assay, although the percentage of indels formation was not calculated [74]. It is important to note that, even though RNP are traditionally preferred over plasmids (DNA) or mRNA, especially when working with primary cells, purified Cas9 proteins are costly and are an important limitation for many laboratories. Therefore, plasmids carrying both Cas9 sequences and mRNA as an all-in-one plasmid are also feasible. All-in-one plasmids have shown high DSB at the AAVS1 [75] and Rosa26 [76] locus in mammalian cells, supporting the idea that low-cost alternatives can be successfully implemented. (See lines 258-273)
- Figure 2: The figure needs to be more described in the legend. What do HD, HNH, etc., stand for? It is unclear.
Answer. We sincerely appreciate this comment, which refers to Figure 1, and confirm we have clarified each portion as follows: Note that the RuvC endonuclease domain comprises three segments: RuvC I, II, and III, while the HNH catalytic domain comprises only one segment. Likewise, three alpha-helical domains (HD) at the REC lobe are placed and primarily responsible for nucleic acid binding. Finally, L-I and L-II are key linkers that aid the connection between RuvC and HNH. (See lines 85-88)
- Table 1: The use of Cas9 is crucial in KI approaches. The table is incomplete and needs to be further described in the text. What are the advantages for each Cas9? Which Cas9 is the best for KI? miCas9 is not included in the table.
Answer. We have updated the table as requested by the reviewer and have included a description as follows: Currently, several modifications on Cas9 have been developed, as summarized in Table 1, and they have been modified to primarily decrease the potential off-target effect of the Cas9 proteins, although some have also been considering their fusion to key regulators of the HDR pathway to increase the knock-in efficiencies. On the other hand, an important concern in CRISPR/Cas9-based therapies is the potential Cas9-mediated unwanted DSB, which can result in genotoxic effects; thus, the use of low off-target producing Cas9 proteins such as Cas9 nickase (nCas9), which require two RNP for inducing DSB, is often chosen over wild-type Cas9 [25]. Likewise, high fidelity Cas9 proteins, mostly containing point mutations to reduce the affinity of the Cas9 protein to the DNA, thus decreasing off-target effects, have been developed, although these genetic modifications can also decrease the on-target cutting [26]. Site-directed mutagenesis was used by Vakulskas et al., 20218 to introduce an R691A at the REC lobe. Interestingly, authors found a lower off-target effect and retained on-target DNA cutting, as observed with wild-type Cas9 [26], providing evidence of a scarless novel Cas9 protein. Even though the selection of the particular variant will depend on the primary goal of the investigator, a variant involving high-fidelity DSB to increase HDR is highly desired. We cover high-fidelity Cas9 in section 4.1.2 of this review paper. (See lines 101-117).
Regarding miCas9, we have included in Table 1 the term HDR-Cas9 as those proteins in which Cas9 has been fused to proteins able to interact with HDR components. Consistently, we have mentioned miCas9 as a promising HDR-Cas9 described in the table legend. References were placed accordingly (See Table 1).
- NHEJ paragraph: There are too many details that are difficult to follow. Is it necessary to understand the current strategies used to improve KI?
Answer. Since we are referring to several proteins involved in the NHEJ pathway as commonest targets for increasing HDR in CRISPR/Cas9 strategies, we consider the current detailed description helpful not only to understand where a potential target is placed but also how downstream NHEJ steps are affected upon inhibition of those proteins. Consistently, we have decided to keep this portion as currently stated.
- 1.2: What is the goal of this paragraph? Can the design of RNA guides improve the KI? It is not clear.
Answer. An increase in Cas9-mediated DNA cutting efficiency will improve the HDR outcome. Since that efficiency depends on the stability between Cas9 and DNA, which is primarily determined by the sgRNA sequence, we have included this paragraph to point it out the parameters that should be considered while designing sgRNA. To clarify this point more, we have included the following: Designing sgRNA is critical for successfully Cas9-mediated DNA cutting. Thus, choosing good sgRNA candidates is critical to favor the interactions between Cas9 and the targeted DNA through base complementary, achieving high on-target cutting [7, 30], which is critical for activating repair mechanisms to induce HDR. (See lines 336-339).
- 1.3. The design of the donor is important to improve KI (Size of arms, size of the donor…). It lacks some details in this review.
Answer. We agree with the reviewer regarding the importance of the donor template as a key factor for improving HDR. However, a very comprehensive review paper in this regard was recently published by Shakirova et al. 2023 that collects up-to-date information in this regard, so we decided to highlight some general considerations (…Conventionally, dsDNA donor templates are preferred to insert larger than 100 nt sequences, while ssDNA is commonly used when less than 100 nt needs to be inserted…) and encourage readers to access to Shakirova’s paper, for avoiding overlapping between both papers.
- Cell type effect: It would be appreciated to have a table of the best strategies to improve HDR efficiency for each cell type.
Answer. We have generated Table 2 to display the current outcomes for each HDR enhancer together with the cell involved in each study so that the reader can see the KI efficiency according to specific cells used, Cas9 variants used, as well as the direct or indirect HDR stimulators, and loci targeted. (Table 2 starts at line 462)
- Conclusion: It would be useful to comment on the disadvantages of some approaches to improve HDR. What are the other consequences on the genome when an NHEJ inhibitor is used?
Answer. We have commented in this regard as follows: For instance, NHEJ is a well-conserved pathway responsible for generating immunoglobin class switch recombination where key components such as Ku70/Ku80, DNA ligase IV, XRCC4, γH2AX, 53BP1, MDC1, and Mre11–Rad50–NBS1 are involved [128, 129]; therefore assessing the impact on the downregulation of these NHEJ components by sh- or si-RNA rely on fundamental concerns that should be address. Likewise, the loss of some NHEJ has been associated with genome instability [130]. Therefore, the potential genotoxic effect of NHEJ protein modulation remains to be uncovered. (See lines 478-484).
Reviewer 2 Report
Comments and Suggestions for Authors
Dear Authors, you present a well-written and highly interesting manuscript. I only have some minor points that should be addressed.
L43/44. Please rephrase this sentence - as NHEJ or HDR depends on the cell cycle.
Table 1: Please add - besides the pros - also the cons of new CRISPR/Cas variants (especially the overall gene editing efficencies and off-targets are of great interest). Approaches that make the editing more precise often reduce overall efficiency. Also consider points from NHEJ/HDR editing in animals and plants e.g. CPF1 mediated HDR in livestock works much worse compared to conventional Cas9 - however it is more precise.
Please provide a Table or Figure with current applications. Perhaps you just could highlight the highest amounts of gene-editings, longest/shortest HDR templates, editings in livestocks, editings in plants,... that have been performed. This is of great interest for the reader.
Author Response
Reviewer 2. Dear Authors, you present a well-written and highly interesting manuscript. I only have some minor points that should be addressed.
- L43/44. Please rephrase this sentence - as NHEJ or HDR depends on the cell cycle.
Answer. We highly appreciate this comment, and have rephrased the sentence as requested by the reviewer: NHEJ is predominantly activated in mammalian cells compared to HDR [10, 11]. Consequently, DBS are more susceptible to be repaired through random indels formation by the activation of the NHEJ pathway, which can be used to induce loss of gene expression (knock-out) rather than specific DNA sequence insertions (knock-in), which is mediated by HDR. Although the CRISPR/Cas9 system is meaningly used for correcting point mutations or inserting expression cassettes via HDR-dependent knock-in [1, 12], knock-out has also been assessed as a therapeutical option in human diseases [13]. (See lines 43-50)
- Table 1: Please add - besides the pros - also the cons of new CRISPR/Cas variants (especially the overall gene editing efficiencies and off-targets are of great interest). Approaches that make the editing more precise often reduce overall efficiency. Also consider points from NHEJ/HDR editing in animals and plants e.g. CPF1 mediated HDR in livestock works much worse compared to conventional Cas9 - however it is more precise.
Answer. We agree with the reviewer and have added the pros and cons of the enlisted Cas9 proteins (See Table 1). Since the HDR efficiency depends on several factors (targeted locus, delivery platform, cell model, etc.), we have decided to enlist the knock-in efficiency of these proteins compared to the wild type in a new table (Table 2), as requested by another reviewer. Even though Cpf1 (Cas12) has gained popularity for genome editing in recent years, this review paper aims to discuss the current alternatives for improving HDR in the specific context of the CRISPR/Cas9 system, so we preferred to limit the discussion only to Cas9, as specified in the title of the paper; thus, we will not include Cpf1 (Cas12).
- Please provide a Table or Figure with current applications. Perhaps you just could highlight the highest amounts of gene-editings, longest/shortest HDR templates, editings in livestocks, editings in plants,... that have been performed. This is of great interest for the reader.
Answer. As per requested by one reviewer, we have included Table 2, which summarizes the major findings in CRISPR/Cas9-based strategies in terms of genome editing efficiency (Knock-in) and explicitly mentioned that donor template considerations were recently covered by Shakirova et al. 2023 (See lines 361-362). Finally, the biomedical applications of the CRISPR/Cas9 system are the major scope of this paper. Thus, as described in the introduction, we did not include any information related to biotechnological applications other than those associated with enhancing HDR in mammalian cells. (See lines 66-69). We appreciate the reviewer’s understanding.
Reviewer 3 Report
Comments and Suggestions for Authors
Andres Felipe Leal and colleagues have submitted a manuscript reviewing the strategies to increase CRISPR-Cas9-based knock-in strategies. Overall, the manuscript reads well and discusses an important field of research that will have significant implications for understanding gene therapy and genome editing. I have the following suggestion for the authors to consider in a revised version-
1. Please rephrase lines 43-49. The sentence is extremely long and disjointed, considering breaking it into specific points.
2. In Table 1, please incorporate different Cas9 available from other bacterial species and their characteristic, For example, FnCas9.
3. In fig-2, please consider elaborating on how single-stranded donors (ssODNs) are used for Homology directed repair (HDR) via BRCA1-mediated single-stranded annealing as compared to double-stranded (ds) donors that are used as templates during homologous recombination (HR) which is BRCA2-dependent. Please elaborate on this section 3.2 and revise the figure accordingly. Although the authors have elaborated briefly in section 4.1.3, I suggest making these sections coherent with Fig-2.
4. In section 4.1.3, symmetrical/asymmetrical homologous recombination arms are known to impact efficiency, the donor template is also important. ssODNs are more efficient than double-stranded donors as SSA is more predominant than HR. This was instrumental in increasing Cas9-based knock-in efficiency for genetic variant generation (refer to PMID-37713444)
5. Section 4.2.3-Consider referencing the work done in enhancing Cas9-based kncockin using Lig4-KO cell line. (PMID-30209399)
6. Discuss the work of Cas9-chimeric fusion, for example-Cas9-RC, a Cas9 fused with eRad18 and CtIP (PMID-37713292).
Comments on the Quality of English LanguageNone
Author Response
Reviewer 3. Andres Felipe Leal and colleagues have submitted a manuscript reviewing the strategies to increase CRISPR-Cas9-based knock-in strategies. Overall, the manuscript reads well and discusses an important field of research that will have significant implications for understanding gene therapy and genome editing. I have the following suggestion for the authors to consider in a revised version-
- Please rephrase lines 43-49. The sentence is extremely long and disjointed, considering breaking it into specific points.
Answer. We have clarified this sentence as requested by the reviewer as follows: NHEJ is predominantly activated in mammalian cells compared to HDR [10, 11]. Consequently, DBS is more susceptible to be repaired through random indels formation by the activation of the NHEJ pathway, which can be used to induce loss of gene expression (knock-out) rather than specific DNA sequence insertions (knock-in), which is mediated by HDR. Although the CRISPR/Cas9 system is meaningly used for correcting point mutations or inserting expression cassettes via HDR-dependent knock-in [1, 12], knock-out has also been assessed as a therapeutical option in human diseases [13]. (See lines 43-50)
- In Table 1, please incorporate different Cas9 available from other bacterial species and their characteristic, For example, FnCas9.
Answer. Table 1 has been changed as requested by two reviewers. In such a table, we have decided to keep current SpCas9 proteins since the main goal of this paper is to review the current alternatives of the CRISPR/SpCas9 system. We have clarified this point as follows: In this review, we detail the molecular mechanisms of the CRISPR/Cas9 system when using Cas9 from Streptococcus pyogenes (SpCas9; hereinafter referred to as Cas9) for knock-in approaches and explore the current strategies attempted to increase their efficiency in mammalian cells. (See lines 66-69).
- In fig-2, please consider elaborating on how single-stranded donors (ssODNs) are used for Homology directed repair (HDR) via BRCA1-mediated single-stranded annealing as compared to double-stranded (ds) donors that are used as templates during homologous recombination (HR) which is BRCA2-dependent. Please elaborate on this section 3.2 and revise the figure accordingly. Although the authors have elaborated briefly in section 4.1.3, I suggest making these sections coherent with Fig-2.
Answer. We have modified Figure 2 by introducing the potential donor templates that can be attempted when working with CRISPR/Cas9; however, we have decided not to detail the full mechanism of BRCA1 in the figure to avoid a hard-to-follow figure. Nonetheless, we have clarified the differences pointed out by reviewer as follows: Several platforms can be used as donor templates, including long single-stranded oligodeoxynucleotide (lssDNA), single-stranded oligode-oxynucleotide (ssODN), double-stranded oligodeoxynucleotide (dsODN), double-stranded DNA (dsDNA) which can be carried through non-viral vectors, as well as DNA templates loaded into viral vectors. For more details related to donor templates and their impact on HDR, we strongly encourage readers to review the paper by Shakirova et al., 2023 [52]. Although differences in the recognition of ssDNA (i.e., BRCA1) and dsDNA (i.e., BRCA2) have been well-documented, in both scenarios, the homologous recombination takes place, leading to precise genome editing in the context of the CRISPR/Cas9 system. (See lines 168-173)
- In section 4.1.3, symmetrical/asymmetrical homologous recombination arms are known to impact efficiency, the donor template is also important. ssODNs are more efficient than double-stranded donors as SSA is more predominant than HR. This was instrumental in increasing Cas9-based knock-in efficiency for genetic variant generation (refer to PMID-37713444) 5. Section 4.2.3-Consider referencing the work done in enhancing Cas9-based knocking using Lig4-KO cell line. (PMID-30209399).
Answer. A very comprehensive review paper in this regard was recently published by Shakirova et al. 2023 that collects up-to-date information about the implications of donor templates in CRISPR/Cas9. To avoid overlapping with Shakirova’s paper, we decided to highlight some general considerations (…Conventionally, dsDNA donor templates are preferred to insert larger than 100 nt sequences, while ssDNA is commonly used when less than 100 nt needs to be inserted…) and encourage readers to access to Shakirova’s paper.
- Discuss the work of Cas9-chimeric fusion, for example-Cas9-RC, a Z and CtIP(PMID-37713292).
Answer. We thank this key comment and agree with the reviewer. We have discussed this point in section 4.1.2 as follows: It is notable that a novel fusion Cas9 protein comprised of eRad18 and CtlP, termed Cas9-RC, was recently tested for increasing HDR through its intra-uterus administration in embryonic mouse brain [86] for inserting an expression cassette carrying mCherry at the Actβ gene. Cas9-RC led to a significant increase of up to 3.7-fold knock-in increase compared to non-fused Cas9 protein. Despite these promising findings, Cas9-RC did not show enhanced knock-in efficiency as compared to Cas9 at the Negr1 locus [86], suggesting that either internal chromatin aspects on the targeted locus and large size Cas9 variants can limit the efficiency of the CRISPR/Cas9 system. (See lines 314-322). These interesting findings were also included in the new Table 2.